# Anti-Influenza Virus Study of Composite Material with MIL-101(Fe)-Adsorbed Favipiravir

**DOI:** 10.3390/molecules27072288

**Published:** 2022-03-31

**Authors:** Mengyuan Xu, Xi Li, Huiying Zheng, Jiehan Chen, Xiaohua Ye, Tiantian Liu

**Affiliations:** School of Public Health, Guangdong Pharmaceutical University, Guangzhou 510310, China; 18762176339@163.com (M.X.); lixi8813@163.com (X.L.); zhyzhy_2021@163.com (H.Z.); chenjie.han@163.com (J.C.); smalltomato@163.com (X.Y.)

**Keywords:** MIL-101(Fe), favipiravir, antibacteria, antiviral

## Abstract

Nanomaterial technology has attracted much attention because of its antibacterial and drug delivery properties, among other applications. Metal-organic frameworks (MOFs) have advantages, such as their pore structure, large specific surface area, open metal sites, and chemical stability, over other nanomaterials, enabling better drug encapsulation and adsorption. In two examples, we used the common pathogenic bacterium *Staphylococcus aureus* and highly infectious influenza A virus. A novel complex MIL-101(Fe)-T705 was formed by synthesizing MOF material MIL-101(Fe) with the drug favipiravir (T-705), and a hot solvent synthesis method was applied to investigate the in vitro antibacterial and antiviral activities. The results showed that MIL-101(Fe)-T705 combined the advantages of nanomaterials and drugs and could inhibit the growth of *Staphylococcus aureus* at a concentration of 0.0032 g/mL. Regarding the inhibition of influenza A virus, MIL-101(Fe)-T705 showed good biosafety at 12, 24, 48, and 72 h in addition to a good antiviral effect at concentrations of 0.1, 0.2, 0.4, 0.8, 1.6, and 3 μg/mL, which were higher than MIL-101(Fe) and T-705.

## 1. Introduction

Influenza is an acute respiratory infection caused by members of the Orthomyxoviridae family. Although three genera of Orthomyxoviridae cause respiratory illness in humans, influenza A virus is the most virulent [1]. Influenza viruses are single-stranded negative-sense RNA viruses that are highly contagious with short incubation periods. The World Health Organization estimates that 1 billion cases occur worldwide each year, including 3 to 5 million severe cases and 290,000–650,000 deaths due to influenza-related respiratory diseases [2]. The current prevalent seasonal influenza viruses mainly include influenza A H1N1, H3N2, and B viruses [3]. The different genetic characteristics of influenza A virus and the antigenicity of hemagglutinin (HA) and neuraminidase (NA) allow the division of influenza viruses into several subtypes. Studies have thus far identified 16 subtypes of influenza A virus HA, denoted as H1–H16, and 9 subtypes of NA, N1–N9 [4]. Several retrospective studies have reported that influenza pandemics occur in cycles of approximately 10–40 years due to antigenic drift [5] while statistics show a total of 4 pandemics caused by influenza viruses [6,7]. The first recorded pandemic began in 1918 and was caused by an influenza A H1N1 virus, killing between 50 and 100 million people worldwide from 1918–1919. The second pandemic began in 1957 and was caused by an influenza A H2N2 virus, containing N2 NA, H2 HA, and polymerase basic (PB) 1 fragments from avian viruses, and 5 other fragments from the 1918 H1N1 virus. This outbreak led to 1.1 million deaths worldwide from 1957–1959. The third pandemic was caused by an influenza A H3N2 virus, which was a human-avian recombinant virus with H3 HA and PB1 fragments from avian viruses and 6 other fragments from the 1957 H2N2 influenza virus. The fourth pandemic was first identified in 2009 and was caused by a swine-derived influenza virus, subsequently named A(H1N1)pdm09, resulting in more than 18,000 deaths of laboratory-confirmed cases.

Drug therapy is important in the fight against influenza virus infection. The mechanisms of action of currently available anti-influenza drugs are limited, so new drugs with novel modes of action are needed. Early licensed anti-influenza drugs included two major classes: alkylamines and NA inhibitors (NAIs), but due to the widespread resistance of influenza viruses to alkylamines, this class of drug is no longer recommended for clinical use [8,9]. Although NAIs are widely used, their efficacy is often reduced due to resistant strains of viruses [10,11]. Polymerase inhibitors are novel anti-influenza drugs that inhibit the influenza virus RNA-dependent RNA polymerase (RdRp) [12]. One of these drugs, favipiravir (T-705), was approved in Japan in 2014 for use against drug-resistant influenza viruses [13]. In addition to influenza viruses, T-705 is effective against a variety of RNA viruses in vitro and in animal models, including severe acute respiratory syndrome coronavirus 2 (SARS-CoV-2) [14,15], Zika virus [16], chikungunya virus [17], Bunya virus [18], and Ebola virus [19].

Bacteria, as the most abundant group of organisms on Earth, are diverse, small, fast-growing, adaptable, and mutable, and are widely present in water, soil, plants, animals, and air. They directly affect our lives, with some having pathogenic effects on humans. Pathogenic bacteria endanger human health in many areas, especially in food safety and health care [20,21]. *Staphylococcus aureus*, which is a major bacterial pathogen, can cause sepsis and bacteremia in the human circulatory system [22,23,24], toxic shock and endocarditis due to toxin release from human organs [25], and pathology caused by surgical contamination and secondary surgical injuries. These negative effects undermine public health and place a heavy burden on economic and health care systems.

With ongoing intensive nanotechnology research, nanomaterials are considered to be the most promising pharmaceutical agents in the 21st century. They not only play an important role in photothermal and antibacterial therapies, but also show excellent antiviral properties. Elimelech et al. [26] reported that the most primitive single-walled carbon nanotubes can interact directly with *Escherichia coli*, causing severe disruption of the cell wall, which leads to bacterial death. Silver nanoparticles have been shown to be novel antibacterial agents that can replace traditional antibiotic drugs [27,28,29,30,31]. Shukla et al. [32] found that zinc oxide nanomaterials have preventive, therapeutic, and neutralizing effects on herpes simplex virus type II. Sarid et al. [33] found that functionalized graphene oxide can block the adsorption of herpes simplex virus type I. Tang et al. [34] found that graphene oxide can rapidly detect both enterovirus 71 and H9N2 influenza virus and disrupt the virus structure. Metal-organic frameworks (MOFs) are materials formed by ligating metal ions or metal clusters as central particles with organic ligands [35,36]. Owing to their large specific surface area and high porosity, adjustable pore size, easy modification, unsaturated metal sites, and biodegradability, they are used in drug encapsulation, bioimaging, biosensing, biocatalysis, and antibacterial applications [37]. The low-toxicity iron analogue MIL-53 (Fe), consisting of a terephthalate anion and a trans chain of an iron (III) octahedron, has a loading capacity of 20 wt% by impregnating the solids in a solution containing n-hexane IBU [38]. Porous MIL-100 (Fe) was used as a nanocarrier for the antitumor drug doxorubicin. The drug was given at a dose of 9 wt% and was released completely after 2 weeks [39]. Ibuprofen loading and release studies of MIL-100 and MIL-101 found that MIL-101 can load 1.4 g of ibuprofen per gram, demonstrating a high drug loading rate and controlled release [40]. Cabrera-Garcia et al. [41] synthesized amino-functionalized MIL-100(Fe)-loaded camptothecin. This nanoplatform released four times more drug under acidic conditions than under normal conditions and effectively improved cellular internalization [42]. The variety of iron-based MOFs available offers many possibilities for achieving appropriate controlled release of various pharmacological molecules [43]. They also have the advantages of high stability, easy preparation, and low cost. Therefore, we attempted to load MIL-101(Fe) with the anti-influenza drug favipiravir (T705) to determine whether it can inhibit *S. aureus* and influenza A viruses and to evaluate the application of MIL-101(Fe)-T705 in antibacterial and antiviral applications.

## 2. Experimental Section

### 2.1. Materials

Dulbecco’s modified eagle medium (DMEM), phosphate-buffered pH 7.4 (1×, 0.25% trypsin-EDTA (1×), and penicillin-streptomycin (100,000 U/mL) were purchased from Gibco (Carlsbad, CA, USA). Fetal bovine serum (FBS) was obtained from Shanghai ExCell Bio (Shanghai, China). Favipiravir (T-705, 98%) was obtained from Shanghai Macklin Bio (Shanghai, China). Terephthalic acid (H_2_BDC), ferric chloride hexahydrate (Fecl_3_·6H_2_O), *N*,*N*-dimethyl formamide (DMF), and chicken erythrocyte were purchased from GBCBIO Technologies (Guangzhou, China). Culture bottles were obtained from Corning Life Sciences Co., Ltd. (New York City, NY, USA). The 96-well plates were purchased from Guangzhou Jet Bio-Filtration Co., Ltd. (Guangzhou, China). Glycerin (99%) was purchased from Aladdin Reagents Ltd. (Shanghai, China). LB broth and nutritional AGAR (NA) medium were obtained from Guangdong Huankai Microbial Technology Co., Ltd. (Guangzhou, China). A hydrothermal synthesis reactor was purchased from Zhengzhou Boke Instrument Equipment Co., Ltd. (Zhengzhou, China). An electric blast and constant temperature drying oven (101-0B) were purchased from Shaoxing Licheng Instrument Technology Co., Ltd. (Shaoxing, China). A ThermoSorvallST16R refrigerated centrifuge and ABI7500 real-time quantitative PCR instrument were purchased from Thermo Fisher Scientific (Waltham, MA, USA).

### 2.2. Synthesis of MIL-101(Fe)

MIL-101(Fe) was synthesized using a solvothermal method according to the literature [42]. In total, 1.714 g of terephthalic acid (H_2_BDC) and 0.824 g of ferric chloride hexahydrate (Fecl_3_·6H_2_O) were dissolved in 15 mL of *N*,*N*-dimethylformamide (DMF), and dispersed uniformly by ultrasound for 15 min. The evenly dispersed solution was placed in a hydrothermal reaction kettle and placed in the oven at 150 °C for 20 h. Then, the reactor temperature was allowed to drop to room temperature. At this point, the supernatant was clear with reddish-brown deposits at the bottom. The supernatant and precipitate were collected, respectively. The precipitate was washed with ethanol and then centrifuged using a high-speed centrifuge (10,000× *g* rpm, 10 min). The process was repeated three times. Finally, after washing, the material was placed in the oven for drying at 110 °C for about 8 h, and the collected product was MIL-101(Fe).

### 2.3. Synthesis of MIL-101(Fe)-T705 Nanocomposites

Similar to the synthesis of MIL-101(Fe), 1.714 g of terephthalic acid (H_2_BDC) and 0.824 g of ferric chloride hexahydrate (Fecl_3_·6H_2_O) were dissolved in 15 mL of DMF (*N*,*N*-dimethylformamide) and sonicated for 15 min to ensure good dispersion. Then, the calculated favipiravir was added and sonication continued for 20 min. All solutions were poured into the reaction kettle and reacted in the oven at 150 °C for 20 h. At this time, the supernatant was clear, and the precipitate was at the bottom. The supernatant and precipitate were collected, respectively. The precipitate was washed with ethanol and then centrifuged using a high-speed centrifuge (10,000× *g* rpm, 10 min). The process was repeated three times. Finally, after washing, the material was placed in the oven for drying at 110 °C for about 8 h, and the collected product was MIL-101(Fe)-T705. The added amount of T-705 in the prepared MIL-101(Fe)-T705 was set at 1.714 g of the total amount of terephthalic acid (H_2_BDC) and 0.824 g of ferric chloride hexahydrate (Fecl_3_·6H_2_O), which was 0.1%, 0.2%, 0.4%, 0.8%, and 1%, respectively. This was named 0.1% MIL-101(Fe)-T705, 0.2% MIL-101(Fe)-T705, 0.4% MIL-101(Fe)-T705, 0.8% MIL-101(Fe)-T705, and 1% MIL-101(Fe)-T705. In total, 2.5, 5.1, 10, 20, and 25 mg of T-705 were added during the synthesis.

### 2.4. Nanomaterial Characterization

The powder X-ray diffraction (XRD, Rigaku Ultima IV, Japan) was analyzed in a 2θ range from 5 to 50° using Cu-Kα radiation. For the Fourier-transform infrared (FT-IR, Bruker ALPHA II, Karlsruhe, Germany) spectra, MIL-101(Fe), MIL-101(Fe)-T705, and T-705 were recorded in the wavelength range of 500–4000 cm^−1^. Thermogravimetric analysis (TGA, NETZSCH STA 2500, Selb, Germany) of MIL-101(Fe)-T705 was set to a target temperature of 800 °C, a heating ramp of 10 °C min^−1^, and an airflow rate of 100 mL min^−1^ under an air atmosphere. The zeta potential and particle size distribution were assessed using a Malvern Zetasizer Nano ZS90 instrument (Malvern, England). X-ray photoelectron spectroscopy (XPS, Thermo fisher Scientific K-Alpha+, USA) was carried out to investigate the chemical compositions, states, and valence band of MIL-101(Fe)-T705 using Al Kα radiation. The morphology, shape, and size of MIL-101(Fe) and MIL-101(Fe)-T705 were examined with a scanning electron microscope (SEM, TESCAN Mira4, Brno, Czech Republic) operated at an accelerating voltage of 200 eV–30 keV. A transmission electron microscope (TEM, Titan G260-300, Hillsboro, OR, USA) was used to determine the internal structure of MIL-101(Fe) and MIL-101(Fe)-T705. The chemical composition of MIL-101(Fe)-T705 was further analyzed using an energy-dispersive X-ray spectrometer (EDS, Zeiss EVO, Oberkochen, Germany). The Brunauer–Emmett–Teller (BET, Micromeritics ASAP 2020 v4.03, Norcross, GA, USA) surface areas and porous structure were evaluated using N_2_ (77.4 K) adsorption-desorption experiments. The UV-vis absorption spectra were measured with a HITACHI U-3010 ultraviolet-visible diffuse reflectance spectrophotometer (Hitachi Limited, Kyoto, Japan) using DMSO as a reference.

### 2.5. Loading and In Vitro Release of Drug

To drawing the standard curve, 1 mg of T-705 was weighed in 1 mL of DMSO to obtain a 1 mg/mL favipiravir storage solution. T-705 solution was diluted to 10, 8 μg, 6 μg, 4, 2, and 0 μg/mL. The absorbance value of the solution was measured with a UV-vis spectrophotometer (λ = 325 nm, HITACHI U-3010, Kyoto, Japan). The standard curve was formulated with the abscissa as the concentration and the ordinate as the absorbance. The absorbance value of the prepared MIL-101(Fe)-T705 supernatant was determined by the UV-fixed wavelength (λ = 325 nm, HITACHI U-3010, Kyoto, Japan). Then, the measured absorbance value was substituted into the obtained T-705 standard curve formula to obtain the concentration value, and substituted into the following formula:(1)$$\;\mathrm{Drug}\;\mathrm{content}\;\left(\mathrm{mg}/\mathrm{mg}\right)=\frac{m_{0}\times \rho \times V\times n}{m_{1}}\times 100\%$$
where $m_{0}$ = T-705 input; $m_{1}$ = total amount of MIL-101(Fe)-T705; *ρ* = the mass concentration of the T-705 supernatant obtained from the standard curve; *V* = supernatant liquid volume; and *n* = ratio of the total supernatant to the measured amount.

The release of the drug T-705 from MIL-101(Fe)-T705 was estimated in PBS solution at 2 different pH values (pH 7.2 and pH 5.5) at 37 °C. Approximately 10 mg of MIL-101(Fe)-T705 were placed in a dialysis bag (MD 34 mm) and slowly shaken in 10 mL of PBS. Subsequently, the PBS solution released from MIL-101(Fe)-T705 at various time points was measured at 325 nm with a UV-vis spectrophotometer (HITACHI U-3010, Kyoto, Japan).

### 2.6. Bacterial Culture

*Staphylococcus aureus* (*S. aureus*) was provided by Xiaohua Ye, Guangdong Pharmaceutical University. After the bacteria reached the logarithmic growth period, the single colony taken from the plates by inoculation loops dissolved in normal saline, and the Maxwell standard turbidimetric tube method was used to obtain 1 × 10^8^ CFU/mL bacterial suspension. The bacterial solution was diluted 100 times to obtain 1 × 10^6^ CFU/mL (CFU: colony-forming unit) of bacterial suspension and stored.

### 2.7. Determination of the Bacterial Survival Rate

The bacterial suspension of 1 × 10^6^ CFU/mL was divided into 6 equal parts, and a series of concentrations (0, 0.0002, 0.0004, 0.0008, 0.0016, and 0.0032 g/mL) of MIL-101(Fe)-T705 were added for shaking on a 37 °C water bath shaker (120× *g* rpm, 24 h). The bacteria survival rate was calculated by measuring OD600 with a microplate analyzer.

### 2.8. Determination of the Minimum Inhibitory Concentration

We used the minimum inhibitory concentration (MIC) and the minimum bactericidal concentration (MBC) as indicators of the antimicrobial ability of MIL-101(Fe)-T705. The smaller the MIC value, the stronger the ability to inhibit bacterial growth, and the smaller the MBC, the stronger the bactericidal ability. In total, 6 EP tubes were taken and coded as No. 1–6. No. 6 was used as a blank control group. No. 1–5 were added to LB medium containing MIL-101(Fe)-T705 (0.0002, 0.0004, 0.0008, 0.0016, and 0.0032 g/mL), respectively. No. 6 was added to pure LB medium. Then, equal amounts of 1 × 10^6^ CFU/mL bacterial suspension were added to each of the 6 FP tubes, shaken well, and placed in a 37 °C shaker (120 r, 24 h). The minimum concentration corresponding to the tube with clarified and transparent culture solution was the minimum inhibitory concentration (MIC) of the nanomaterial against the species of bacteria. The concentration of the material corresponding to the plate with less than five colonies or no bacteria production was the minimum bactericidal concentration (MBC) of the nanomaterial for that species of bacteria.

### 2.9. Determination of the Inhibitory Growth Curve

The inhibition curve of the bacterial growth with different concentrations of MIL-101(Fe)-T705 was examined. The enzyme standard plate was prepared and 1 × 10^8^ CFU/mL of bacteria was added to each well. The LB medium containing MIL-101(Fe)-T705 (0, 0.0002, 0.0004, 0.0008, 0.0016, and 0.0032 g/mL) was added to each well. The final well was zero-adjusted, and only LB medium containing the corresponding concentrations of MIL-101(Fe)-T705 was sterile. The blank control group was set, and the same amount of bacterial liquid without synthetic materials was added to the LB medium. The plate was placed in a shaker for incubation, and its OD_600_ value was measured with a microplate reader every hour. The inhibition growth curve of the bacteria over 24 h was plotted with culture time as the abscissa and the absorbance value at 600 nm as the ordinate.

### 2.10. Cell Culture and Virus Amplification

Madin-Darby canine kidney cells (MDCK) were purchased from the American Type Culture Collection (Manassas, VA, USA). PR8 strain of influenza A virus (A/PR/8/34, H1N1) was provided by Professor Zhou Rong (State Key Laboratory of Respiratory Diseases, Guangzhou, China). MDCK cells was cultured in DMEM containing 1% (*v*/*v*) penicillin/streptomycin and 10% (*v*/*v*) FBS in 25 or 75 cm^2^ cell culture flasks at 37 °C/35 °C, 5% CO_2_, 95% humidity. H1N1 influenza virus was cultured in chicken embryos and stored at −80 °C. The culture medium was exchanged for DMEM containing 1% penicillin/streptomycin, 5% bovine serum albumin (BSA), and TPCK-treated trypsin before infection of MDCK cells. The virus was genetically modified so that it could not infect human cells.

### 2.11. Cytotoxicity Test of MIL-101(Fe)-T705

MDCK cells were inoculated in 96-well plates at a density of 1 × 10^5^ cells/well and incubated at 37 °C with 5% CO_2_ for 24 h. At the end of the culture, the stock solution was aspirated, and different concentrations of MIL-101(Fe)-T705 (0.1, 0.2, 0.4, 0.8, 1.6, and 3 μg/mL) were added to 96-well plates with good cell formation and a standard cell control group was set up (MIL-101(Fe)-T705 was lysed with 0.1% DMSO). The cells were incubated for 12, 24, 48, and 72 h. The absorbance values at OD_490 nm_ were measured with an enzyme marker using the MTT method. The cell viability was calculated at different concentrations of MIL-101(Fe)-T705.

### 2.12. Virus Titer Determination

Viral titers of influenza viruses were measured using the half tissue culture infectious dose (TCID_50_) to measure virulence. The influenza virus was serially diluted with DMEM to 10^−1^–10^−8^. The influenza virus diluted at each concentration was added to 96-well plates with good layers of MDCK cells, 100 μL/well. After incubation in the incubator at 35 °C for 1 h, the mixture was carefully aspirated, and 100 μL of virus maintenance solution were added to continue the incubation, in all 8 replicate wells. Moreover, a standard cell control group was set up. Cell lesions were observed daily, the extent of lesions and the number of wells were recorded, and the results were recorded after the cells were no longer lesioned. Culture wells with less than 50% cell lesions were recorded as non-lesioned wells, and those with more than 50% were recorded as lesioned wells. The TCID_50_ of the virus was calculated according to the Reed–Muench method.

### 2.13. Determination of the Growth Curve of Influenza Virus

MDCK cells were seeded at 1 × 10^6^ cells/well in 6-well plates at 37 °C and 5% CO_2_ for 24 h. After the end of the culture, the culture medium was discarded, and washed with PBS twice. DMEM was used to dilute the virus, with MOI = 0.001. Then, 800 μL of diluted disease venom were added to each well and incubated at 35 °C with 5% CO_2_ for 1 h. The disease venom was sucked out and washed with PBS at the end of the adsorption. Finally, virus maintenance solution with different concentrations of MIL-101(Fe)-T705 was added and incubated in an incubator. The supernatant was collected after 12, 24, 36, 48, and 72 h of culture and stored at −80 °C. After repeated freezing and thawing 3 times, TCID_50_ was determined.

### 2.14. Determination of Fluorescence Quantitative PCR

MDCK cells were inoculated at 1 × 10^5^ cells/well in 24-well plates and incubated at 37 °C with 5% CO_2_ for 24 h. After the culture, the culture medium was discarded and washed twice with PBS. DMEM was used to dilute the virus at MOI = 0.001. In total, 200 μL of the diluted virus were added to each well and incubated for 1 h at 35 °C and 5% CO_2_. MIL-101(Fe)-T705 was diluted with virus maintenance solution to 0.1, 0.2, 0.4, 0.8, 1.6, and 3.0 μg/mL. After the virus was adsorbed, the virus venom was sucked out, and a virus maintenance solution containing different concentrations of MIL-101(Fe)-T705 was added and incubated at 35 °C with 5% CO_2_ for 24 h. After the cells in the 24-well plate were washed with PBS, 250 μL of Trizol reagent were added to each well. The cells were gently blown and left for 5 min before being transferred to the EP tube of an RNA enzyme. Then, 150 μL of chloroform were added to the EP tube, shaken for 15 s, left to stand for 5 min, and centrifuged for 12,000× *g* at 4 °C for 15 min. After centrifugation, the top layer solution was drained into a new RNA-free EP tube, an equal volume of isopropanol solution added, and mixed upside down. After standing at room temperature for 10 min, the EP tube was centrifuged at 4 °C for 5 min. The supernatant was discarded, and the white precipitate was washed with 75% ethanol (DEPC water) to remove the ethanol. Then, the open EP tube was dried until the white precipitate gradually became translucent, and 50 μL of DEPC water were added, gently blown to dissolve it, and stored in a refrigerator at −80 °C for later use. The RT-PCR reaction system is shown in Appendix A.

### 2.15. Statistical Analysis

All experimental results were expressed as the mean ± standard deviation (mean ± SD), and the data were analyzed using Graph Prism 7. The *t*-test was used to compare the experimental and control groups, and *p* < 0.05 was used as the criterion for the significance of differences.

## 3. Results

### 3.1. Characterization of the As-Made MIL-101(Fe)-T705

MIL-101(Fe)-T705 was synthesized using the solvothermal synthesis method. First, terephthalic acid and ferric chloride hexahydrate were reacted in DMF, followed by the addition of the drug favipiravir. The nanocomposites were obtained under high-temperature and high-pressure conditions in the reactor, and a schematic diagram of the whole reaction process is shown in Figure 1.

We compared the UV-Vis absorption spectra of MIL-101(Fe)-T705, MIL-101(Fe), and T-705, and analyzed the preparation of MIL-101(Fe)-T705 at different temperatures and different favipiravir dosages, with dimethyl sulfoxide (DMSO) as the reference material. The absorption peak of MIL-101(Fe)-T705 was around 260 nm, consistent with MIL-101(Fe), and the absorption peak became higher and wider, covering T-705 (Appendix A). Similarly, as shown in Appendix A, the half-peak width of the absorption peak of nanoparticles also gradually shrank and narrowed with the increasing reaction temperature, which can be explained by the fact that at 150 °C, the obtained uniform dispersion of the nanoparticle size was improved. MIL-101(Fe)-T705 had significant absorption in the UV region in the wavelength range of 200–350 nm (Appendix A). At 0.1% MIL-101(Fe)-T705, the half-peak width was the narrowest and, with the increasing concentration of T-705, the half-peak width increased until 0.8% MIL-101(Fe)-T705. The highest absorption peak was seen at this time. However, there was no longer a significant absorption peak at 1% MIL-101(Fe)-T705. This can be explained by the increasing concentration of T-705, which resulted in the pore structure of the loaded MIL-101(Fe) not being uniform. Therefore, we concluded that the 0.8% concentration of T-705 was most suitable for binding with MIL-101(Fe) to become MIL-101(Fe)-T705.

The crystal structures of the samples were investigated using X-ray powder diffraction (XRD). As shown in Figure 2, MIL-101(Fe) showed distinct diffraction characteristic peaks at diffraction angles of 2θ = 8.92°, 9.26°, 15.57°, 18.72°, and 21.91°. From the related literature [44], MIL-101(Fe) has a distinct characteristic peak at 2θ = 5–25°, which matches with the JCPDS card database, indicating that the in-house-synthesized sample was indeed MIL-101(Fe). Meanwhile, the X-ray diffraction pattern of favipiravir (T-705) obtained from the crystal structure database showed that 2θ = 24.11°, 25.02°, 27.87°, 29.58°, 33.11°, 35.04°, 35.61°, 38.12°, and 40.91° of MIL-101(Fe)-T705 corresponded to the T-705 diffraction peaks. Therefore, from the above results, we concluded that MIL-101(Fe) and T-705 were successfully synthesized as MIL-101(Fe)-T705.

The entire range of the Fourier-transform infrared spectroscopy (FT-IR) spectra are divided into 2 regions: 4000–1300 and 1300–600 cm^−1^. The range of 4000–1300 cm^−1^ reflects the characteristic stretching vibrations of functional groups and chemical bonds. Within 1300–600 cm^−1^, in addition to the above, superimposed effects from environmental unknowns and other factors are observed, which is called the fingerprint region. This part is not fully explained, so it is mostly used for comparison. As shown in Figure 3, FT-IR of the T-705 sample showed absorption peaks located at 3200 and 1660 cm^−1^, representing the presence of characteristic peaks of -NH_2_ and C=O, while the characteristic peak at 1057 cm^−1^ is unique to the imidazole group within T-705. T-705 is an aromatic heterocyclic compound due to the presence of the imidazole group, and 4 characteristic peaks of varying intensity unique to aromatic compounds were visible at 1596, 1558, 1465, and 1431 cm^−1^. MIL-101(Fe) showed completely different characteristic peaks, with absorption peaks at 722, 1013, 1275, 1383, 1504, and 1688 cm^−1^. The characteristic peak at 722 cm^−1^ originated mainly from the vibration of the C-H bond in the benzene ring. Further, 1383 and 1504 cm^−1^ could be interpreted as symmetric and asymmetric vibrations of the carboxyl-COOH group, and the characteristic peak at 1688 cm-1 was associated with the presence of C=O bonds in the free carboxyl group, indicating a continuous dicarboxyl linkage [45,46]. In the FT-IR of MIL-101(Fe)-T705, characteristic absorption peaks of MIL-101(Fe) located at 727, 776, 1013, 1280, 1392, 1509, and 1679 cm^−1^, and 1606, 1572, 1509, and 1426 cm^−1^ for T-705 aromatic heterocyclic characteristic peaks, demonstrated the synthesis of MIL-101(Fe)-T705.

The thermogravimetric analysis (TGA) curves of MIL-101(Fe)-T705 in air are shown in Appendix A. MIL-101(Fe)-T705 showed an obvious decomposition stage, with a thermal decomposition rate of approximately 75%. When the temperature was around 300 °C, MIL-101(Fe)-T705 started to decompose, and when the temperature reached 420 °C, the mass no longer changed. At this time, MIL-101(Fe)-T705 had completely decomposed, mainly because the high-temperature condition led to the collapse of the hydroxyl group in the MIL-101(Fe)-T705 skeleton by shedding the skeleton. We also measured the zeta potential (Appendix A) and particle size distribution (Appendix A) of MIL-101(Fe)-T705, and the particle size distribution ranged from 200 to 500 nm, with an average particle size of 377.4 ± 4.89 nm and a polydispersity index of 0.35, which was uniformly dispersed. Generally, as the absolute value of the zeta potential increases, the electrostatic repulsion between particles increases, and the system is less likely to settle and agglomerate, and therefore the stability is improved. The average potential of MIL-101(Fe)-T705 was around 15.8 ± 4.79 mV, and the surface of the nanomaterial was positively charged with a certain stability.

Combined with the above results, we concluded that MIL-101(Fe) was synthesized successfully. Similarly, we also compared the scanning electron microscopy (Figure 4A,B,D,E) and fluoroscopic electron microscopy (Figure 4C,F) of MIL-101(Fe) and MIL-101(Fe)-T705 in the same field. The appearance of MIL-101(Fe) aggregates after T-705 loading can be seen in the figure. Transmission electron microscopy also showed that favipiravir was encased in the originally hollow MIL-101(Fe). Here, we suspect that the drug exists not only inside MIL-101(Fe) but also residues on its surface. The distribution of C, O, and Fe represent the establishment of the MIL-101(Fe) skeleton centered on trivalent iron while F and N indicate the favipiravir distribution (Figure 4G,H). This is consistent with the above XRD results and further proves the successful synthesis of MIL-101(Fe)-T705 nanocomplexes.

To estimate the specific surface area, we used the Brunner–Emmet–Teller (BET) method to analyze the pore volumes and pore diameters of MIL-101(Fe) and MIL-101(Fe)-T705 to obtain N_2_ adsorption–desorption isotherms. MIL-101(Fe) exhibited a typical type IV N_2_ adsorption–desorption curve, indicating that it is a porous material [47]. In contrast, the N_2_ adsorption–desorption curve of MIL-101(Fe)-T705 is flat (Figure 5A). Combined with the corresponding pore size distribution graphs (Figure 5B) and Table 1, the specific surface area of MIL-101(Fe) was observed to be 199.7194 m^2^ g^−1^ with a pore size of 2.33771 nm, and the specific surface area of MIL-101(Fe)-T705 is 116.7785 m^2^ g^−1^ with a pore size of 1.5195 nm. Therefore, the specific surface area and pore volume of MIL-101(Fe)-T705 decreased after the addition of T-705. This was probably caused by T-705 filling the MIL-101 (Fe) spaces.

To further investigate the surface chemical composition and chemical valence of the complex, the elements contained in the MIL-101(Fe)-T705 composite and their chemical composition were analyzed by X-ray photoelectron spectroscopy (XPS) detection. Figure 6A shows the full spectrum of MIL-101(Fe)-T705, showing that MIL-101(Fe)-T705 was mainly composed of the elements C, O, Fe, Cl Cl, F (Figure 6A), and N. Figure 6B–F shows O 1s, C 1s, Fe 2p, F 1s, and N 1s. The 2 characteristic peaks in O 1s, with binding energies of 532.62 and 532.43 eV (Figure 6B), corresponded to C=O in T-705 and Fe-O in MIL-101(Fe), respectively [48,49]. The high resolution of C1s was divided into 3 peaks (Figure 6C), with the characteristic peak at 285.53 eV corresponding to C-N in T-705 [50], and the 2 characteristic peaks at 284.05 and 288.33 eV attributed to the benzene ring and carboxylic acid groups on the organic ligand terephthalic acid in MIL-101(Fe), respectively [51]. In the high-resolution spectrum of Fe 2p, the binding energies were located at 710.36 and 726.13 eV (Figure 6D), corresponding to the Fe 2p1/2 and Fe 2p3/2 peaks of Fe(III), respectively [52]. The characteristic peaks with binding energies of 713.11 and 717.33 eV confirmed the presence of trivalent iron ions in the free state in the material [53]. In addition, due to the low F and N content, energy-dispersive X-ray spectroscopy has a deeper detection depth than XPS. Therefore, when the element contents to be measured are less than 5%, XPS may not be reliable for accurate detection (Figure 6E,F).

The concentration of T-705 in MIL-101(Fe)-T705 supernatant was obtained by substituting the absorbance value of MIL-101(Fe)-T705 supernatant into the formula of the T-705 standard curve (y = 0.0223x − 0.4496, R^2^ = 0.9152). Then, the other variables were substituted into Equation (1), and the drug content of MIL-101(Fe)-T705 was calculated to be 27.03%. In the in vitro drug release experiment, the nanocomposite loaded with T-705 was immersed in PBS and gently shaken. Subsequently, we studied the T-705 release curves of MIL-101(Fe)-T705 at pH 7.4 and 5.5. As shown in Figure 7A, MIL-101(Fe)-T705 demonstrated the pH response release process. When pH was 7.4, the release amount of T-705 was 53.90% after 16 h, and when pH was 5.5, the release amount of T-705 was 65.27%. However, considering the water-soluble characteristics of favipiravir, we believe that the release amount of T-705 is not only related to the release from the pore but also from the favipiravir on the surface of MIL-101 (Fe)-T705.

### 3.2. Antibacterial Properties of MIL-101(Fe)-T705

The concentration of nanomaterials seriously affects the properties of nanomaterials, and thus we analyzed the effect of different concentrations on bacterial survival. As shown in Figure 8A, favipiravir (T-705) did not have a significant inhibitory effect on Gram-positive *S. aureus*. Conversely, with increasing dosages of MIL-101(Fe) and MIL-101(Fe)-T705, the inhibitory effect on *S. aureus* increased, and the synthetic MIL-101(Fe)-T705 was significantly more inhibitory than MIL-101(Fe). The survival rate of *S. aureus* was approximately 70% after exposure to 0.0008 g/mL of MIL-101(Fe)-T705, and approximately 25% after treatment with 0.0016 g/mL of MIL-101(Fe)-T705. However, when the concentration was increased to 0.0032 g/mL, the survival rate of *S. aureus* was zero. Therefore, it can be concluded that MIL-101(Fe)-T705 has some antibacterial activity, with an optimal concentration of 0.0032 g/mL.

The minimum inhibitory concentration (MIC) and minimum bactericidal concentration (MBC) for *S. aureus* were investigated at different concentrations of MIL-101(Fe)-T705, and the results are shown in Figure 8B,C. The concentration of MIL-101(Fe)-T705 corresponding to the clarified and translucent Eppendorf tube (EP) of the bacterial culture represents the MIC of the treatment. The MIC and MBC of MIL-101(Fe)-T705 were 0.0008 and 0.0032 g/mL, respectively.

Figure 8D shows the 24 h growth inhibition curves of different MIL-101(Fe)-T705 against *S. aureus*. Untreated *S. aureus* entered the logarithmic growth phase after 5 h and the plateau phase after incubation for up to 15 h. When cultures of *S. aureus* were treated with 0.0002 g/mL MIL-101(Fe)-T705, the growth inhibition curve was similar to that of untreated *S. aureus*. However, as the concentration of MIL-101(Fe)-T705 increased, the growth inhibition curve gradually leveled off, particularly when the concentrations of MIL-101(Fe)-T705 were 0.0008 and 0.0016 g/mL, as the growth curve plateaued after 15 h, with no further bacterial growth. Furthermore, at a concentration of 0.0032 g/mL MIL-101(Fe)-T705, complete inhibition of bacterial growth was observed.

The results shown in Figure 8A indicate that T-705 has no intrinsic antibacterial activity. Existing studies have also shown that the antibacterial activity of MOFs is due to the presence of metal cations in the framework [54]. Therefore, we believe that the antibacterial mechanism of MIL-101(Fe)-T705 is as follows: the release of Fe^3+^ in MIL-101(Fe) disrupts the metabolism of bacteria and prevents bacterial reproduction. According to the above characterization results, the surface of MIL-101(Fe)-T705 itself is positively charged, enabling binding to negatively charged bacteria, which destroys their structure and results in bacterial cell death. Further studies are required to validate this.

### 3.3. Biocompatibility Assessments

The most important aspect of nanomaterials in their application to biological systems is good biocompatibility. Therefore, we investigated the cytotoxicity of MIL-101(Fe)-T705 in MDCK cells. The results are shown in Figure 9. After co-culture of MIL-101(Fe)-T705 with MDCK cells for 12, 24, 48, and 72 h, differences in survival rate were not statistically significant compared with untreated control MDCK cells, demonstrating that MIL-101(Fe)-T705 was not cytotoxic and therefore has great potential in the study of biological systems. Second, to exclude the effect of DMSO, we also tested the cytotoxicity of 0.1% DMSO (Appendix A), demonstrating that there were no significant differences in the viability of MDCK cells after 0.1% DMSO treatment at 12, 24, 48, and 72 h.

### 3.4. Antiviral Properties of MIL-101(Fe)-T705

Virus entry into host cells mainly involves the processes of adsorption, fusion, cell entry, replication, and release. Common nanomaterials in the antiviral field are mainly focused on direct inactivation of the virus, hindering adsorption and thus invasion of the virus, interfering with virus replication, and preventing the exit of progeny virus from the cell [55]. Influenza A virus H1N1 has eight RNA fragments, seven of which surround the eighth fragment and replicate intracellularly as a whole during the entire virus transmission process [56]. Favipiravir (T-705), which is an RdRp inhibitor, acts mainly at different stages of influenza virus infection and replication, inhibiting viral RdRp by terminating RNA strand extension at the doping site [57]. Therefore, we investigated the blocking effect of MIL-101(Fe)-T705 on H1N1 adsorption into cells. MDCK cells were adsorbed with MIL-101(Fe)-T705, MIL-101(Fe), or T-705 after infection with influenza virus, and the supernatant was collected for median tissue culture infective dose (TCID50) assay. As shown in Figure 10A, when cells were incubated with MIL-101(Fe)-T705 for 1 h followed by an influenza virus infection, the virus titer decreased significantly. Although T-705 and MIL-101(Fe) had slight inhibitory effects on influenza virus (Figure 10B,C), these were not statistically significant. Importantly, MIL-101(Fe)-T705 had a significantly greater inhibitory effect at each dose compared with T-705 alone and MIL-101(Fe), with these differences becoming more apparent with increasing concentrations (Figure 10D). Therefore, we suggest that the mode of action of MIL-101(Fe)-T705 is two-fold. The first mechanism lies in impeding virus adsorption and entry into host cells by MIL-101(Fe) occupying the receptor on the cell membrane that recognizes virus particles while in the second mechanism, T-705 is phosphorylated by cellular enzymes to its active form T-705-RTP, which hinders replication of progeny virus [58], thus achieving the observed antiviral effect.

Following this, we determined the viral titers in the supernatant by PCR assay after pre-absorption of MIL-101(Fe)-T705. The obtained Ct values were compared with previously obtained TCID_50_ (Figure 11A). The results were consistent with those shown in Figure 9A. MIL-101(Fe)-T705 showed a significant inhibitory effect on influenza virus replication. Meanwhile, we also measured the viral titer growth curves of influenza viruses treated with different concentrations of MIL-101(Fe)-T705. As shown in Figure 11B, there were no significant differences in the viral titers between treatment groups at 24 h post-infection, although these groups demonstrated lower titers than untreated controls. However, at MIL-101(Fe)-T705 concentrations of 0.1, 0.2, and 0.4 μg/mL, the virus titer increased from 24 h to reach similar titers to the control group. Meanwhile, when the concentration of MIL-101(Fe)-T705 was 0.8, 1.6, or 3.0 µg/mL, the virus titers did not change significantly, indicating that these concentrations of MIL-101(Fe)-T705 could inhibit influenza virus replication infection.

## 4. Conclusions

Starting from the antimicrobial-loaded drug performance of nanomaterials, we combined the nanomaterial MIL-101(Fe) with the drug favipiravir (T-705) via the solvent thermal method to form a new nanocomposite material MIL-101(Fe)-T705. Taking the common pathogenic bacteria *S. aureus* as an example, the in vitro antibacterial comparison of MIL-101(Fe), T-705, and MIL-101(Fe)-T705 revealed that T-705 had no antibacterial properties. MIL-101(Fe)-T705 inhibited the growth of *S. aureus* at a concentration of 0.0032 g/mL with the help of the antibacterial properties of the nanomaterial itself. As an anti-influenza drug, the main mechanism of action of T-705 lies in disrupting the replication of influenza virus. Inspired by this, we subjected MDCK cells pre-treated with MIL-101(Fe)-T705 to in vitro infection, with an effect on influenza A (H1N1) virus, and the results were surprising: MIL-101(Fe)-T705 showed a good antiviral effect and low cytotoxicity. Furthermore, the antiviral effect of MIL-101(Fe)-T705 was better than that of T-705 and MIL-101(Fe) at the same concentrations. Taken together with the intrinsic antibacterial and antiviral properties of MIL-101(Fe), we believe that MIL-101(Fe) may occupy the site on the cell membrane recognized by influenza virus, thus preventing the virus from attaching and entering the cell. Following this, T-705 is released into the cell to exert anti-influenza effects and inhibit virus replication. In addition to influenza virus, T-705 can inhibit a variety of viruses, indicating important possibilities for future research of nanomaterials combined with antiviral drugs. However, the specific mechanism of action needs further study.

## Figures and Tables

**Figure 1 molecules-27-02288-f001:**
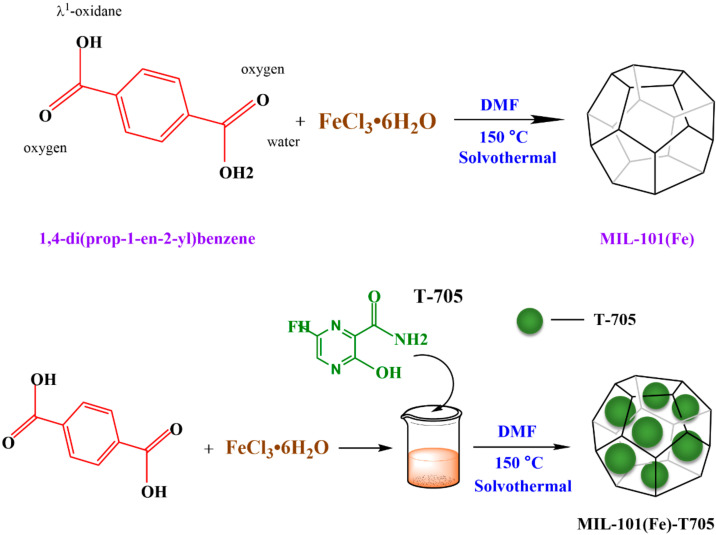
Synthetic schematic diagram of MIL-101(Fe) and MIL-101(Fe)-T705.

**Figure 2 molecules-27-02288-f002:**
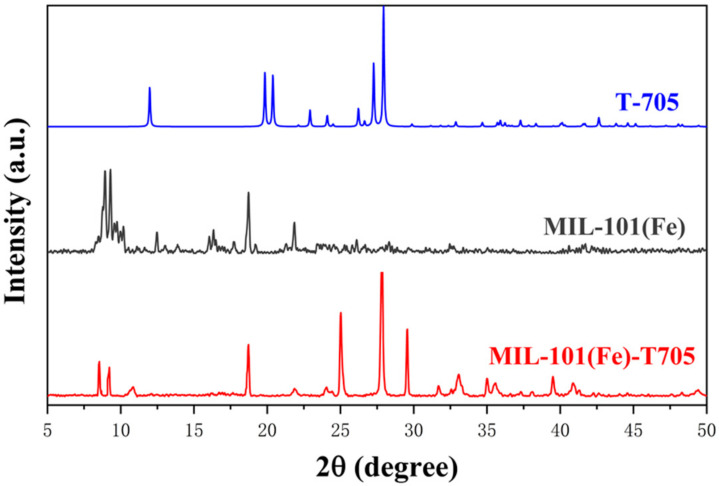
XRD patterns of the as-prepared samples.

**Figure 3 molecules-27-02288-f003:**
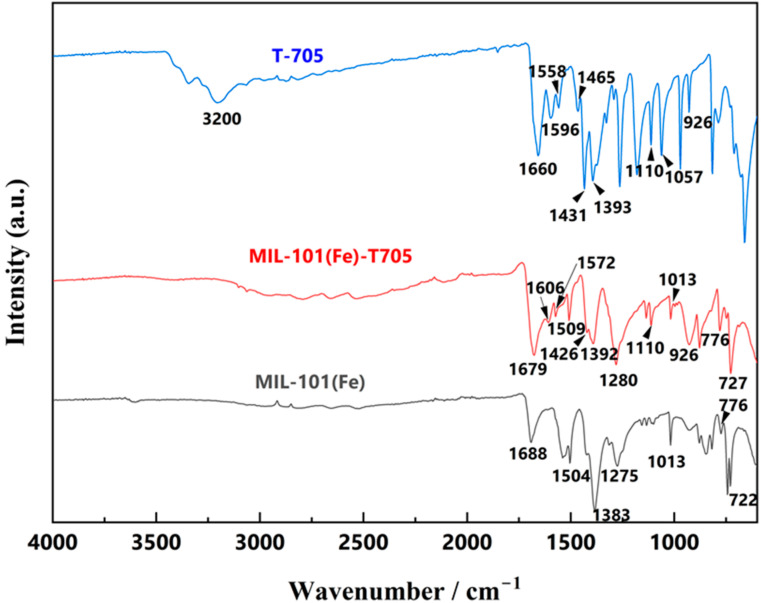
FT-IR spectra of the as-synthesized powders.

**Figure 4 molecules-27-02288-f004:**
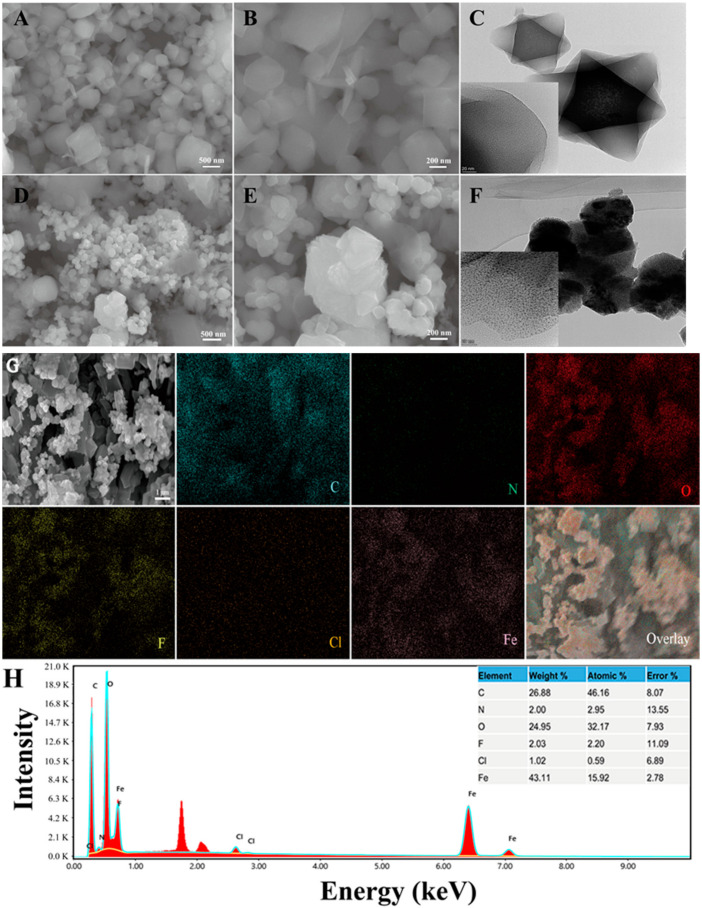
Scanning electron microscopy (SEM) (**A**,**B**) images of MIL-101(Fe) and MIL-101(Fe)-T705 (**D**,**E**). Transmission electron microscopy (TEM) (**C**) images of MIL-101(Fe) and MIL-101(Fe)-T705 (**F**). (**G**) SEM image and energy-dispersive X-ray spectroscopy (EDS) for elemental color mappings (C, N, O, F, Cl, Fe) and (**H**) the EDS spectrum of the MIL-101(Fe)-T705.

**Figure 5 molecules-27-02288-f005:**
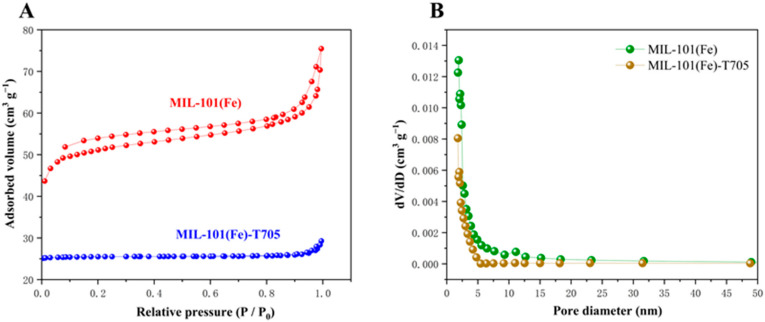
Nitrogen adsorption–desorption isotherms (**A**) and the corresponding pore size distribution (**B**).

**Figure 6 molecules-27-02288-f006:**
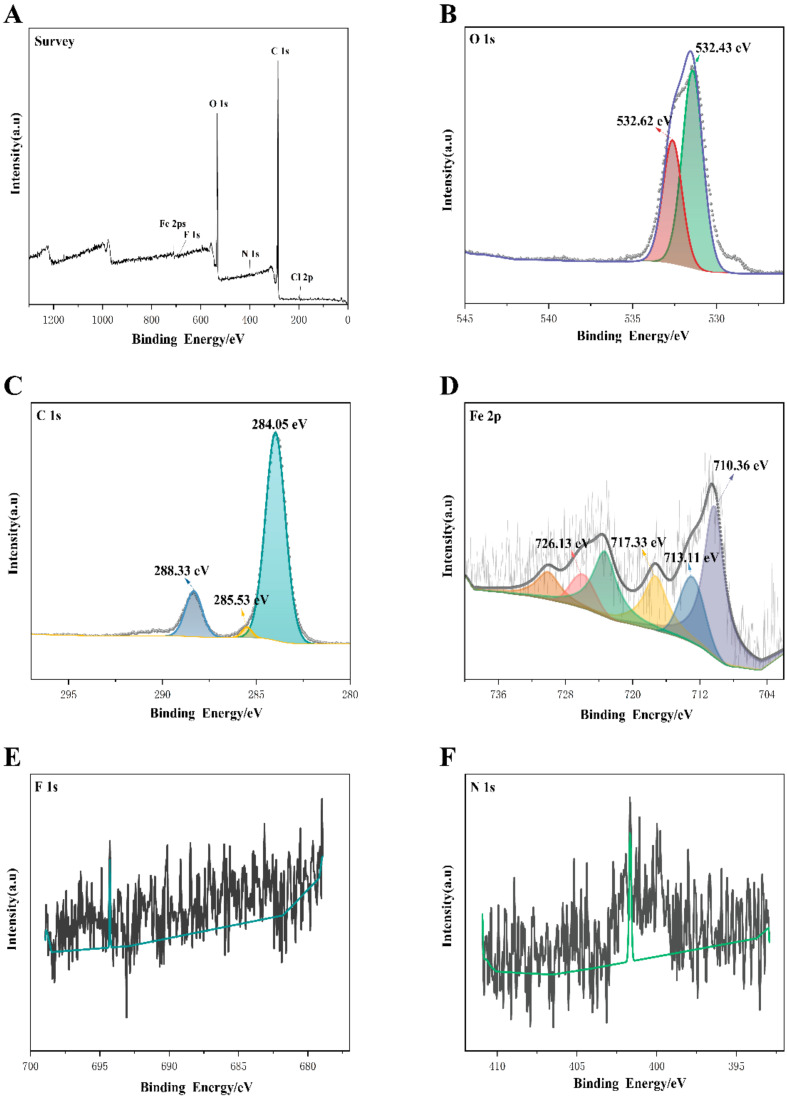
X-ray photoelectron spectroscopy spectra of MIL-101(Fe)-T705 (**A**) survey; (**B**) O 1s; (**C**) C 1s; (**D**) Fe 2p; (**E**) F 1s; (**F**) N 1s.

**Figure 7 molecules-27-02288-f007:**
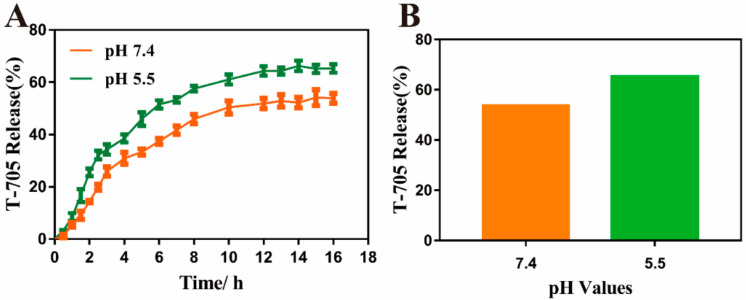
(**A**) The release curve of MIL-101(Fe)-T705 under different pH values. (**B**) The total release of T-705.

**Figure 8 molecules-27-02288-f008:**
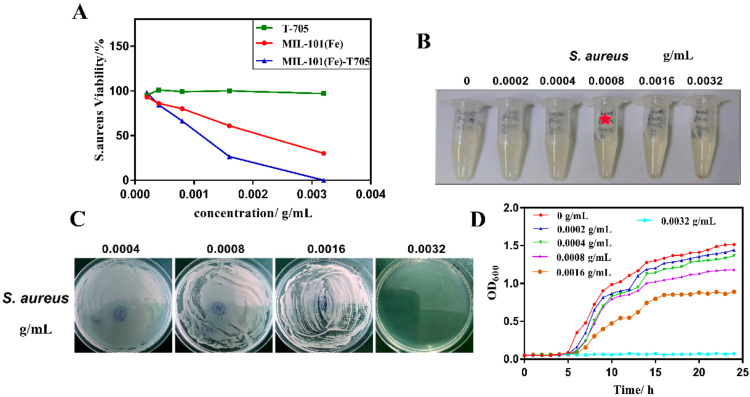
Antibacterial application in vitro. (**A**) Bacterial survival rate of *S. aureus* treated with different concentrations of T-705, MIL-101(Fe), and MIL-101(Fe)-T705. (**B**) Minimum inhibitory concentration of *S. aureus* after treatment with different concentrations of MIL-101(Fe)-T705. The red star means minimum inhibitory concentration of *S. aureus.* (**C**) Minimum bactericidal concentration of *S. aureus* after treatment with different concentrations of MIL-101(Fe)-T705. (**D**) Twenty-four-hour growth curves of *S. aureus* treated with different concentrations MIL-101(Fe)-T705.

**Figure 9 molecules-27-02288-f009:**
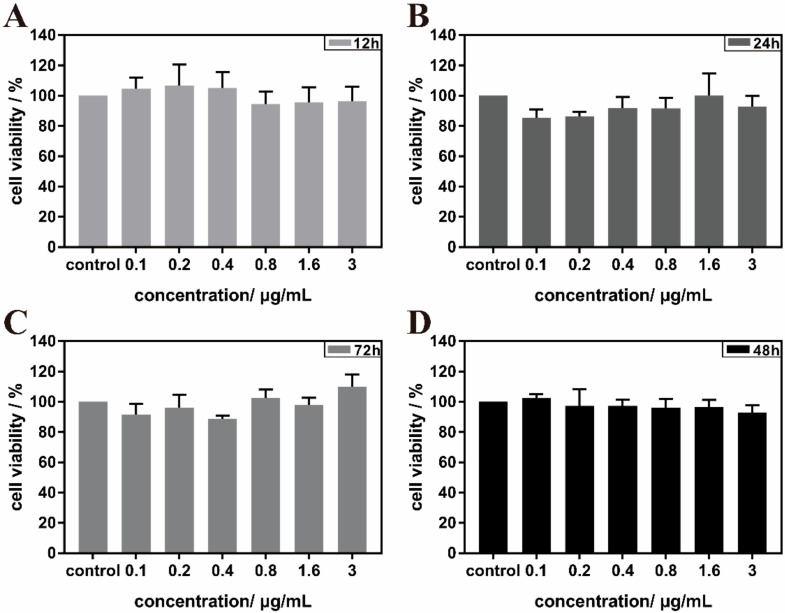
Survival rate of MIL-101(Fe)-T705-treated MDCK cells after 12 h (**A**), 24 h (**B**), 48 h (**D**), and 72 h (**C**).

**Figure 10 molecules-27-02288-f010:**
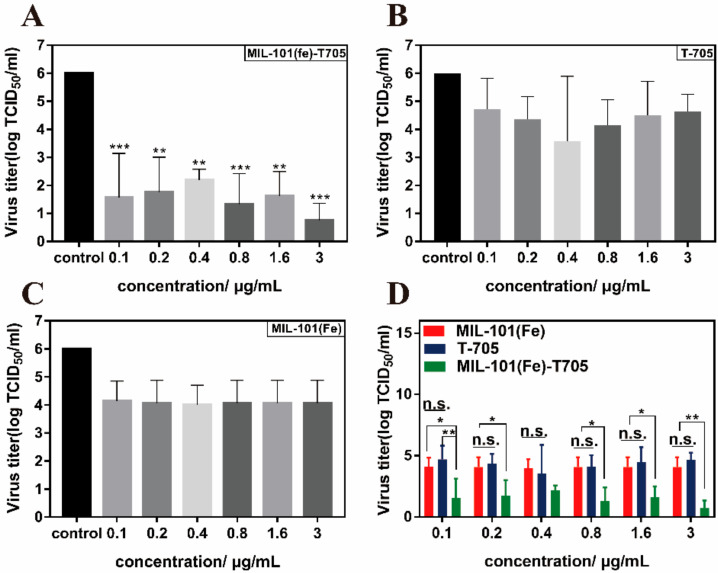
Changes in the titer of influenza virus. (**A**) Determination of H1N1 virus titer following treatment with varying concentrations of MIL-101 (Fe)-T705. (**B**) Determination of H1N1 virus T705. (**C**) Determination of H1N1 virus titers following treatment with varying concentrations of MIL-101(Fe). (**D**) Comparison of virus titers following treatment with varying concentrations of MIL-101(Fe)-T705, T-705, or MIL-101(Fe). * *p* < 0.05, ** *p* < 0.01, *** *p* < 0.001, and n.s. represents no significance. Data were mean ± SD.

**Figure 11 molecules-27-02288-f011:**
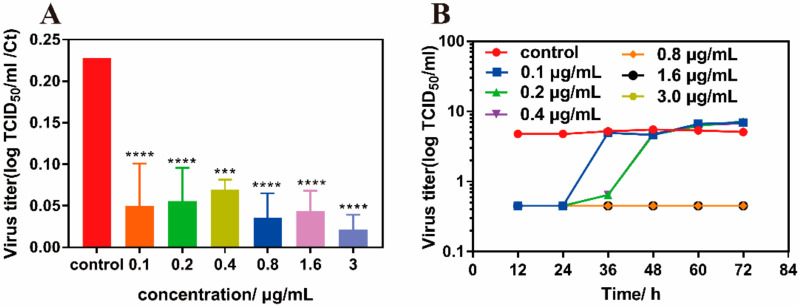
(**A**) Ratio of the Ct value to TCID50 following prophylactic treatment with MIL-101(Fe)-T705 against H1N1 virus infection. (**B**) Growth curves of influenza virus titers in MDCK cells treated with MIL-101(Fe)-T705 at different concentrations for 12, 24, 36, 48, 60, and 72 h were observed. *** *p* < 0.001, **** *p* < 0.0001. Data were mean ± SD.

**Table 1 molecules-27-02288-t001:** Specific surface area, pore diameter, and pore volume of MIL-101(Fe) and MIL-101(Fe)-T705.

Samples	Surface Area (m^2^ g^−1^)	Pore Diameter (nm)	Pore Volume (cm^3^ g^−1^)
MIL-101(Fe)	199.7194	2.33771	0.116722
MIL-101(Fe)-T705	116.7785	1.5159	0.008778

## Data Availability

Not applicable.

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
