# Peer review of "Anti-Influenza Virus Study of Composite Material with MIL-101(Fe)-Adsorbed Favipiravir"

_molecules, 2022, doi:10.3390/molecules27072288_

Round 1

Reviewer 1 Report

In this work, the author reported MIL-101(Fe)-T705  which is prepared by the hydrothermal in situ synthesis using MIL-101(Fe) and drug fabipiravir (T-705). The author demonstrated that MIL-101(Fe)-T705 showed good antiviral utility under the premise of ensuring biosafety safety. The antiviral effect of MIL-101(Fe)-T705 506 was better than that of T-705 and MIL-101(Fe) at the same concentration setting. 

I found the antiviral study is well performed. However, If I see the PXRD and SEM analysis, purity of the material used for the study is a big question. The peak at 17.5 2 theta in Fig. 2 (red) not explained. The first two peaks between 5-10 2 theta are almost disappeared in red line. This clearly signify the material is a physical mixture of drugs and MIL. When I look at SEM picture Fig. 4 C-F, I have the same feeling. Almost no surface area in BET analysis after the incorporation of drug into MIL. 

I would like the author to improve this or provide a valid reason.

Reviewer 2 Report

The manuscript „Anti-influenza virus study of composite material with MIL-101(Fe)-adsorbed fabipiravir” is just reasonable technical study of low scientific value. From the scientific point of view the manuscript presents the next „me too” study. Lot of work was done, but the study is just one of the plethora studies on MOF drug loading. In the introduction the authors mentioned only ibuprofen and camptothecin as two successfully loaded drugs. Please check tons of literature data – there are many review articles on this topic.

What was the reason to load MOF with fabipiravir? What was the intended route of administration? Once successfully loaded into MOFs, should favipiravir (marked as slightly soluble in water) release from the MOFs? Have you any data on drug release?

327-333

What is the reason to test thermal degradation of the MIL-101(Fe)-T705? I think that more interesting issue is degradation in physiological conditions.

341-342

Synthesized MOFs in Figs 4A and 4B do not resemble MOF as presented in Fig. 1. The same with drug-loaded MOFs.

Are you sure that SEM images show six-hole stereo structure of MOF ???? How it is possible to assess internal MOF structure as field of view of 4A image is of ~12um??? Single MOFs are hardly seen at magnification used.

Moreover, as Figs 4A-4B shows separate MOF particles, Figs 4E-4F show one, big agglomerate of drug-loaded MOFs, size of several micrometers. Such agglomeration was intentional?

343-344 „from the SEM of MIL-101(Fe)-T705, it is clear that fapiravir is firmly adsorbed within the six-hole structure in a spherical form.”

It is not clear. Please explain where the spherical form of fabipiravir is presented in SEM images.

358-359 „MIL-101(Fe)-T705 exhibited a typical Type IV nitrogen adsorption-desorption curve, indicating that it is a porous material”

Something is wrong. MIL-101(Fe)-T705 is drug loaded and is still porous?

Figure 5.

Pore size distribution is unreadable. Could you restrict x axis to 0-50 nm range?

Table 1B.

Pore diameter of MIL-101(Fe) is 2.33771 nm and after fabipiravir loading (MIL 101(Fe)-T705) it increased to 20.20013 nm? How is it possible?

387-388 „In addition, the XPS high-resolution spectra of F1s and N1s further indicate the presence of nitrogen and fluorine elements in MIL-101(Fe)-T705.”

Please comment.

420-426

In general, MOF antibacterial properties are nothing surprising – see review article by Wyszogrodzka et al. (Drug Discov Today. 2016 Jun;21(6):1009-18. doi: 10.1016/j.drudis.2016.04.009).

500-501

What is the meaning of the sentence “MIL-101(Fe)-T705 inhibited the growth of gold glucose at a concentration of 0.0032 g/ml with the help of the antimicrobial property of the 501 nanomaterial itself.”. “gold glucose” appears here in conclusions for the first time.

Language issues

Please check manuscript carefully regarding language issues. For example, in introduction past and present tenses are mixed. Another example can be section 2.3 full of strange language constructs and mistakes.

62-69

Very long and, in consequence, unreadable sentence. Please simplify.

Some examples of simple mistakes - it seems that authors did not even use simple spell-checker:

266 „3. RESUITS”

331 „changes, At this time, MIL-101(Fe)-T705 has been …”

203-204 „Influenza A virus strain PR8 (A/PR/8/34,H1N1) was gifts”

When specifying materials, software and laboratory equipment, please, provide full data concerning manufacturer, including city and country, e.g.: (Sigma-Aldrich, Saint-Louis, MO, USA), (Shimadzu, Kyoto, Japan).

Round 2

Reviewer 1 Report

Author provided suitable modifications on the previous version and therefore I am happy to accept this work for publication

Reviewer 2 Report

Regarding my issue #1 and author’s response #1, see the review article by He et al. (Acta Pharm Sin B, 2021 Aug;11(8):2362-2395. doi: 10.1016/j.apsb.2021.03.019.) – there are plenty of advanced studies regarding MOF applications. I can repeat the sentence from the first round of the review: „The manuscript … is just reasonable technical study of low scientific value”. The study is an introduction to further works. Moreover, there are still several unsupported claims/conclusions. The authors tried to improve the manuscript but the manuscript is still sloppy. Therefore, I cannot recommend it for publication.

Some issues are listed below:

The authors changed XRD pattern for MIL-101(Fe)-T705 in the resubmitted version. What was the reason? The authors repeated synthesis of MIL-101(Fe)-T705? New SEM and TEM images were performed also on newly synthesized samples? Or on the original ones? But FT-IR, EDS results are exactly the same (Figures 3, 4). I’m confused.

Figure 7.

The total released amount of T-705 is ca. 60%. In my opinion the authors have no proof that the released T-705 released from the MOF pores. Maybe slightly soluble in water drug was released mainly from the surface?

Figure 4

Particular sub-images are not referenced in the manuscript (only Figure 4 as a whole). SEM results are not described/discussed in the manuscript!

360-365

“The distribution of C, O, and Fe represented the establishment of the MIL-101(Fe) skeleton centered on trivalent iron, while the distribution of F and N indicated that the favipiravir particles were in the form of spheres. The distribution of elements such as F and N indicated that the favipiravir particles were embedded on the skeleton.” - I assume that the passage refers to Figure 4G. The field of view of images in Figure 4G is ~10um. Referring to Figure 4G, please explain in details what are the premises that “distribution of F and N indicated that the favipiravir particles were in the form of spheres”.

377

What is the meaning of the sentence “In contrast, the MIL-101(Fe)-T705 curve remained almost unchanged.” Which curve remained unchanged? You mean that adsorption/desorption isotherms are almost identical? Please be precise.

168-171

“Substitute the OD value into the following formula” - What is OD? How should I substitute OD value into the formula (line 171)???

414

What is „the drug loading rate” and how it was measured?

-------------------------------------------------------------------

The authors claim that the language has been polished. I’m not a native English speaker, but I can still find strange language constructions. Some examples are listed below:

125-126

“10,000r centrifugation for 10min was washed with ethanol and repeat 3 times, dried in the oven at 110°C for about 8h …” - Centrifugation is just technique/process/method and cannot be washed and dried (for example see: https://www.fishersci.se/se/en/scientific-products/centrifuge-guide/centrifugation-theory.html)

147-149

“Thermogravimetric analysis (TGA, NETZSCH STA 2500, Germany), target temperature of 800°C, a heating ramp of 10°C min-1 and an air flow rate of 100ml min-1 under air atmosphere.” – It’s not a sentence (where is the predicate?)

175

“The release amount of T-705 in MIL-101(Fe)-T705 was estimated in PBS solution.” – What is the meaning of this sentence??. Release is a process. Such sentences should not appear in scientific publication.

373-375

“To estimate the specific surface area, we used the BrunnerEmmetTeller (BET) method …, and the obtained N2 adsorption/desorption isotherms as shown in Figure 5.” – are shown?

376

“a typical type IV Nadsorption-desorption curve” – what is nadsorption?

414-417

All paragraph is italic formatted. Why?

505

„with theose”

-----------------------------------------

There are some „instruction like” sentences – are they taken from manuals?:

Section 2.2

119-122

„MIL-101(Fe) was synthesized by a solvothermal method … “ but „Then pour into the reaction kettle, oven 150°C reaction for 20 h.” – this sentence is even somewhat strange.

123

“Cool to room temperature and take out."

124

„Then carefully aspirate the supernatant and collect the precipitate.”

Section 2.5

“Take the supernatant of the prepared MIL-101(Fe)-T705 and measure the OD value. Substitute the OD value into the following formula.” – The reader should take the supernatant and measure the OD value?

Section 2.9, 207-209

„Examine the inhibition curve of bacterial growth with different concentrations of MIL-101(Fe)-T705. Prepare the enzyme standard plate by adding 1×108 CFU/mL of bacteria in each well.”

Section 2.13

“Dilute the virus with DMEM at MOI=0.001. Add 800 μL virus solution per well, incubate at 35°C, 5% CO2  for 1 h. At the end of adsorption, aspirate the virus solution, wash once with PBS, add virus maintenance solution containing different concentrations of MIL-101(Fe)-T705, and continue incubation in the incubator.”
